# Vapor Composition and Vaporization Thermodynamics of 1-Ethyl-3-methylimidazolium Hexafluorophosphate Ionic Liquid

Anatoliy M. Dunaev [1,*], Vladimir B. Motalov [1], Mikhail A. Korobov [1], Dmitrii Govorov [2], Victor V. Aleksandriiskii [3] and Lev S. Kudin [1]

1    Department of Physics, Ivanovo State University of Chemistry and Technology, Ivanovo 153000, Russia
2    Department of Chemistry, University of Cincinnati, Cincinnati, OH 210172, USA
3    Department of Chemistry and Technology of High-Molecular Compounds, Ivanovo State University of Chemistry and Technology, Ivanovo 153000, Russia
*    Correspondence: amdunaev@ro.ru

**Abstract:** The processes of the sublimation and thermal decomposition of the 1-ethyl-3-methyli midazolium hexafluorophosphate ionic liquid (EMImPF$_6$) were studied by a complex approach including Knudsen effusion mass spectrometry, IR and NMR spectroscopy, and quantum chemical calculations. It was established that the vapor over the liquid phase primarily consists of decomposition products under equilibrium conditions. Otherwise, the neutral ion pairs are the only vapor components under Langmuir conditions. To identify the nature of the decomposition products, an experiment on the distillation of the ionic liquid was performed and the collected distillate was analyzed. It was revealed by the IR and NMR spectroscopy that EMImPF$_6$ decomposes to substituted imidazole-2-ylidene (C$_6$N$_2$H$_{10}$PF$_5$) and HF. The measured vapor pressure of C$_6$N$_2$H$_{10}$PF$_5$ reveals a very low activity of the decomposition products ($<10^{-4}$) in the liquid phase. The absence of a significant accumulation of decomposition products in the condensed phase makes it possible to determine the enthalpy of sublimation of the ionic liquid assuming its unchanged activity. The thermodynamics of the EMImPF$_6$ sublimation was studied by Knudsen effusion mass spectrometry. The formation enthalpy of EMImPF$_6$ in the ideal gas state was found from a combination of the sublimation enthalpy and formation enthalpy of the ionic liquid in the condensed state. The obtained value is in good agreement with those calculated by quantum chemical methods.

**Keywords:** ionic liquids; imidazolium; hexafluorophosphate; Knudsen effusion mass spectrometry; quantum chemical calculations; thermodynamics; evaporation; decomposition; vapor pressure





## 1. Introduction

The vaporization of ionic liquids (ILs) with highly electronegative anions under high vacuum conditions is accompanied by the partial decomposition of ILs with the formation of substituted imidazole-2-ylidenes. In particular, such decomposition products were clearly identified for ILs with the BF$_4^-$ [1–3], PF$_6^-$ [4], dicyanamide, thiocyanate, tri-cyanomethanide, and vinylogous dicyanamide anions [5]. The only exception is the work of Volpe et al. [6], where the formation of alkylimidazoles was postulated for 1-butyl-3-methylimidazolium hexafluorophosphate (BMImPF$_6$), apparently erroneously; see [3] for details. As reported in literature [1–3,6], the competing evaporation/decomposition results in complex vapor composition with the fractions of neutral ion pairs (NIPs) and decomposition products dependent on evaporation conditions such as the IL-volume-to-surface ratio [1,4] or the evaporation-to-effusion-area ratio [2,3,6]. The ambiguity of the factors that determine evaporation/decomposition competition calls for new research on this topic. In our recent work [3], a new approach to analyzing the complex vapor composition was developed on the example of 1-butyl-3-methylimidazolium tetrafluoroborate (BMImBF$_4$).

It includes a mass spectrometric investigation of IL under different evaporation conditions: (1) in equilibrium inside an effusion cell (EC), (2) in nonequilibrium over an open IL surface (OS), and (3) in intermediate conditions in an open cell (OC). In this work, we apply the same approach for 1-ethyl-3-methylimidazolium hexafluorophosphate (EMImPF$_6$), being a representative of alkyl-imidazolium ILs with the PF$_6^-$ anion. This IL is expected to decompose according to the reaction through formation Arduengo-type N-heterocyclic carbene [4]

$$EMImPF_{6,\,s} = C_6N_2H_{10}PF_{5,\,g} + HF,\,_g,$$

where C$_6$N$_2$H$_{10}$PF$_5$ is 1-ethyl-3-methylimidazolium-2-pentafluorophosphate (Figure 1).

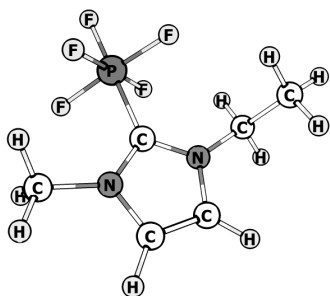

**Figure 1.** Structure of the 1-ethyl-3-methylimidazolium-2-pentafluorophosphate.

The complex vapor composition makes it difficult to determine the thermodynamic characteristics of the vaporization. To date, the only investigation of the vaporization thermodynamics of EMImPF$_6$ is the work of Zaitsau et al. [7] performed by the quartz crystal microbalance (QCM) method. The thermodynamic quantities in [7] were measured under nonequilibrium Langmuir conditions without any control of the gas phase composition. The goal of this work is to carry out a thorough investigation of the vapor species over EMImPF$_6$ and to determine the thermodynamic parameters of its sublimation.

## 2. Materials and Methods

EMImPF$_6$ is in the solid state at room temperature ($T_m$ = 333 K [8]). IR spectra of solid samples (99% purity, Abcr GmbH, Karlsruhe, Germany) were obtained by a Bruker Tensor 27 FTIR Spectrometer in 4000–400 cm$^{-1}$ spectral range with 1 cm$^{-1}$ resolution.

1H, 13C, and 31P NMR spectra in DMSO-d6 at room temperature were recorded by a Bruker Avance III 500 (Bruker AXS, Madison, WI, USA) spectrometer with 5 mm TBI 1H/31P/D-BB z-GRD sensor. Detailed description of the apparatus was given elsewhere [2].

The Knudsen effusion mass spectrometry (KEMS) experiments were carried out in the electron ionization mode ($E$ = 40 eV) with the use of a magnetic sector apparatus MI1201. Evaporation of the samples was performed from molybdenum Knudsen cells. Two cells with different ratios of evaporation area to effusion area, about 600 (EC-I) and about 200 (EC-II), were used. Sample heating was controlled by the temperature regulator OWEN TRM101 coupled with a tungsten–rhenium thermocouple. The melting point of lithium (453.69 K) was used to calibrate the temperature with an accuracy of ±3 K. A detailed description of the mass spectrometer is given elsewhere [9–11].

To study the decomposition products, IL was evaporated during 1.5 h at $T$ = 630 K under high vacuum and the distillate was collected and analyzed. The experiment was carried out until the sample in effusion cell was completely distilled.

Computations of the thermochemical data of the studied compounds were performed in the Gaussian 16 environment [12] using the composite G4 method [13]. A detailed description of the method is given elsewhere [14]. The thermodynamic functions (enthalpy increment $H°(T)–H°(0)$ and reduced Gibbs energy $φ°(T)$) of the compounds in the ideal gas state were computed on a basis of the molecular parameters obtained at B3LYP/6-31G(2df,p) level of theory in rigid rotor–harmonic oscillator approximation using StatThermo software [15]. The uncertainties in $φ°$ $(T)$ were assessed as 2% according to [16].

## 3. Results and Discussion

### 3.1. IR Spectroscopy

The IR spectrum of the initial IL (Figure 2) is in good agreement with the literature [17]. The spectrum of the residue after KEMS is generally close to that of the initial IL; a few minor new peaks appeared, indicating the presence of the traces of the decomposition products in the condensed phase. In the spectrum of the distillate, these peaks become more distinct, and some additional signals appeared. The intensity of the bands related to the C-H stretching vibrations in the aromatic ring (3200 and 3050 $cm^{-1}$) is much lower in the distillate than in the initial sample. The intensity of the peak at 915 $cm^{-1}$ corresponding to the out-of-plane C2-H vibration and 1250 $cm^{-1}$ corresponding to the rocking C2-H vibration becomes weaker in the distillate. This fact points out a substitution of the hydrogen atom in the C2 position in the IL during the formation of decomposition products.

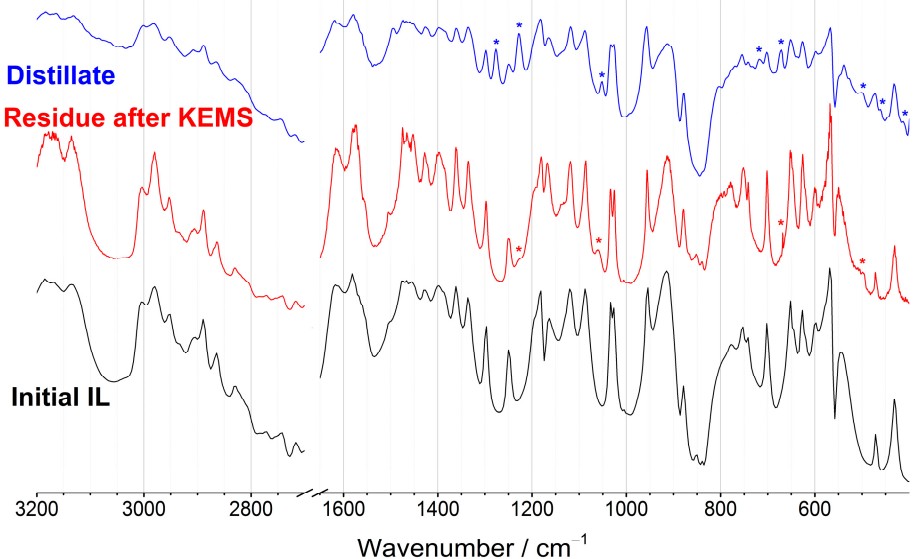

**Figure 2.** IR-spectra of the studied compounds. Peaks corresponding to decomposition products are marked by the asterisk.

### 3.2. NMR-Analysis

The $^{1}$H NMR spectra (Figure 3) of the initial IL and the residue after KEMS are identical indicating the absence of the decomposition products in the solid phase in a significant amount. However, the spectrum of the distillate clearly differs from the previous ones, showing the presence of the decomposition products. The signals of the C4-H, C5-H, and C8-H protons are shifted upfield by ~0.09 ppm. A similar behavior was observed for the BMImBF$_4$ ionic liquid [4]. In contrast, the peaks of the C6-H protons are shifted downfield by 0.12 ppm. A new small singlet signal at 1.24 ppm appeared. The considerable decrease of the C2-H proton intensity is a distinctive feature of the distillate spectrum indicating the formation of 1-ethyl-3-methylimidazolium-2-pentafluorophosphate ($C_6N_2H_{10}PF_5$) by the substitution of hydrogen in the C2 position. The relative amount of $C_6N_2H_{10}PF_5$ was assessed by the integration of the C4-H and C5-H signals to be ~90%. None of the traces of any imidazoles were found in the distillate.

The $^{13}$C and $^{31}$P NMR spectra (Figures 4 and 5) confirmed the $C_6N_2H_{10}PF_5$ formation. In the $^{13}$C spectrum, the C4, C7, and C8 signals are shifted downfield by 0.4, 0.7, and 1 ppm, respectively. The maximum shift of 2.0 ppm downfield is observed for the C6 atom. The C5 signal is shifted by 0.7 ppm upfield in the distillate spectrum. It should be noted that the C4 and C5 signals in the distillate spectrum are doublets, appearing due to the spin–spin interaction of these atoms with the PF$_5$ group in the C2 position. This phenomenon was also observed for the homologous BMImPF$_6$ IL [4]. The C2 signal almost completely disappeared. The changes in the $^{31}$P spectra are more evident. We observed not only the shifts of the signals but also the changing of the number of peaks from septet to

sextet. This clearly indicates the transformation of the $PF_6^-$ anion into the $PF_5$ group in the substituted imidazole-2-ylidene.

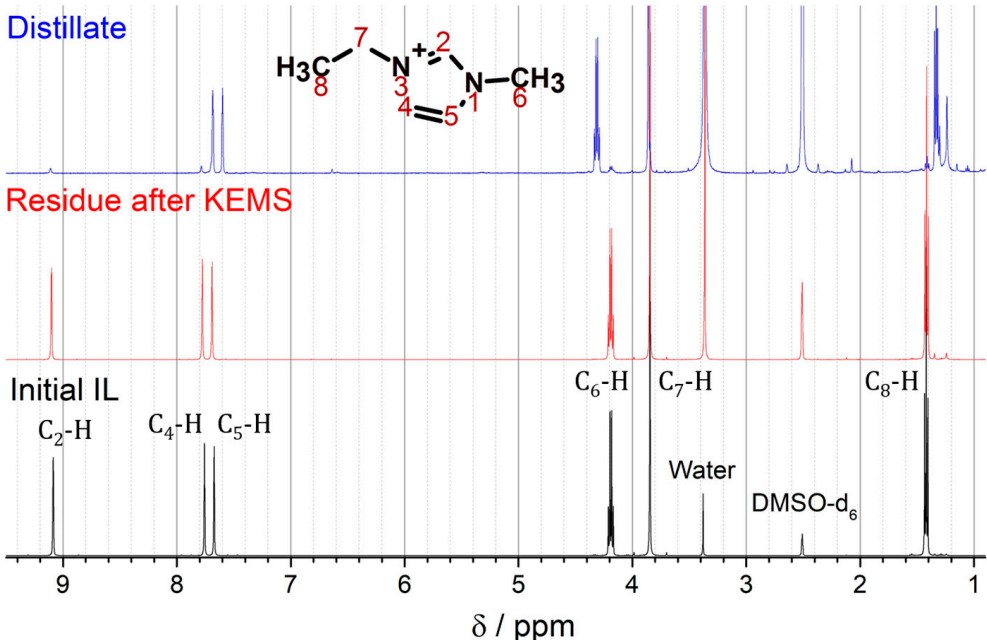

**Figure 3.** $^1$H NMR spectra of studied compounds.

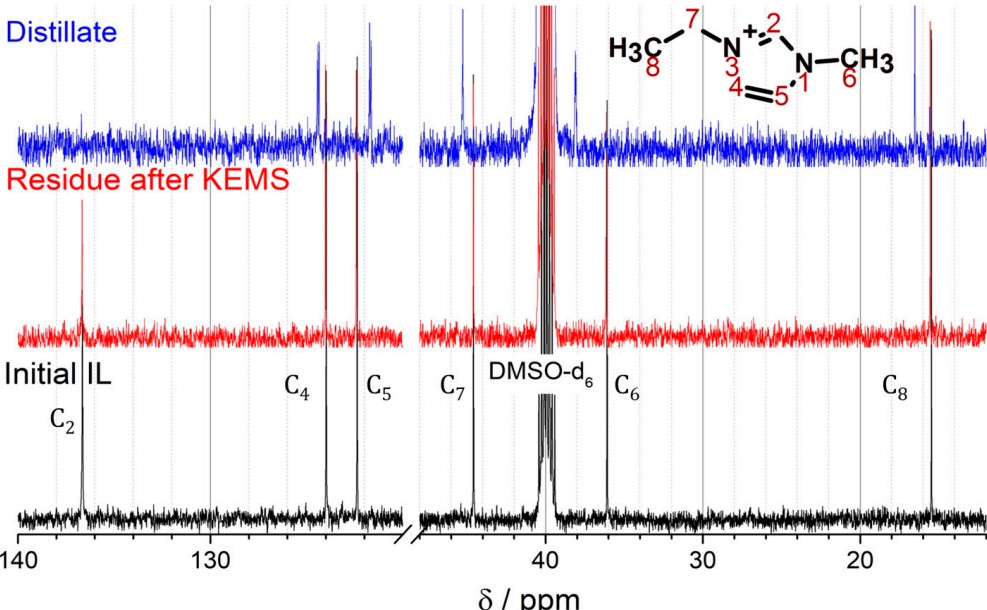

**Figure 4.** $^{13}$C NMR spectra of studied compounds.

### 3.3. KEMS

The background subtracted electron ionization mass spectra of the most intensive ions under different evaporation conditions are shown in Figure 6. The full-scale mass spectra of the $EMImPF_6$ can be found in Supplementary Materials (Figures S1–S4). One can see that the spectra under OC conditions are temperature-independent. Under equilibrium conditions, a slight temperature trend of the increasing intensity of ions corresponding to the fragmentation of $C_6N_2H_{10}PF_5$ (see below) was revealed. During the analysis of the mass spectra presented in Figures 6 and 7, the intensity of the parent cation $C_6N_2H_{11}^+$ was corrected by the intensity of the second isotope of the ion $C_6N_2H_{10}^+$ since the latter makes a significant contribution to the peak with $m/z = 111$. The ion with $m/z = 82$ is

dominating under equilibrium conditions, while, under OS and OC conditions, the parent cation with $m/z = 111$ is the most intensive. In all spectra except OS, the ions corresponding to the fragmentation of $C_6N_2H_{10}PF_5$ are present. The peak with $m/z = 217$ corresponds to fluorine detachment ($C_6N_2H_{10}PF_4^+$) and that with $m/z = 169$ corresponds to a cleavage of $C_6N_2H_{10}PF_4^+$ onto $C_2H_5F$ and $C_4N_2H_5PF_3^+$. All these ions are fingerprints of the formation of the substituted imidazole-2-ylidene. Similar ions were observed in the homologous $BMImPF_6$ in our work [4].

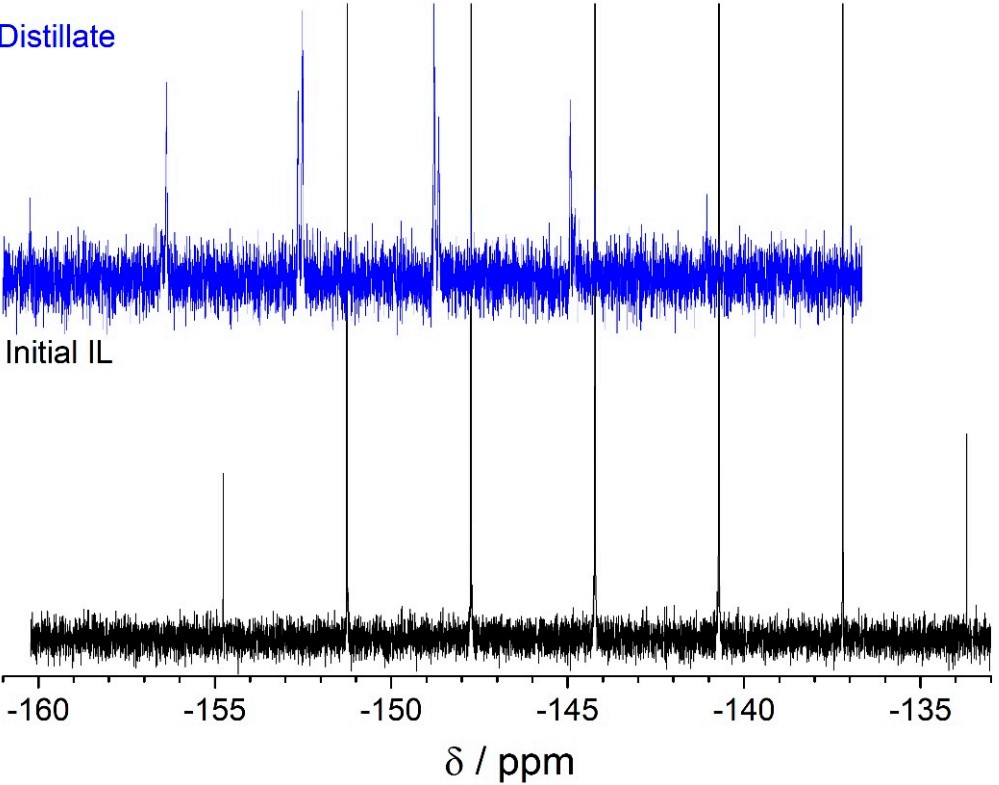

**Figure 5.** $^{31}P$ NMR spectra of studied compounds.

Under OC conditions, two interesting ions appear in the spectrum. One of them with $m/z = 130$ corresponds to the fluorine attachment to the parent cation. The same kind of ions were previously registered in similar $BMImPF_6$ and $BMImBF_4$ ILs [2–4,18]. The scheme of the formation of this ion is given in [18]. Another ion with $m/z = 367$ registered at 524 K (0.2% of the parent cation intensity) corresponds to the formula $C_2A^+$, where C is the parent cation and A is the $PF_6^-$ anion. This fact indicates the formation of a small amount of NIP dimer. Previously, dimerization was observed only for the ILs evaporated without decomposition [19].

The mass spectrum recorded under OS conditions contains only ions with $m/z = 82$, 110, and 111. As it is shown earlier [2], evaporation in the nonequilibrium condition suppresses the decomposition processes, and the OS spectrum corresponds to that of NIP.

The mass spectrum of the distillate (Figure 7) corresponding to that of $C_6N_2H_{10}PF_5$ contains the following major ions (relative to intensity of ion with $m/z = 82$): 82 (100%), 217 (60%), 110 (50%), 169 (17%), 107 (15%), and 81 (12%). The full-scale mass spectrum of the distillate is given in Supplementary Materials (Figure S5).

Ionization efficiency curves (IEC) for the most intensive ions were measured under different conditions and temperatures (Figure 8). The energy scale was calibrated by the ionization energy of $H_2O$ (*IE* = 12.62 eV [20]). Appearance energies (*AE*) of the ions were determined by the vanishing current method (Table 1).

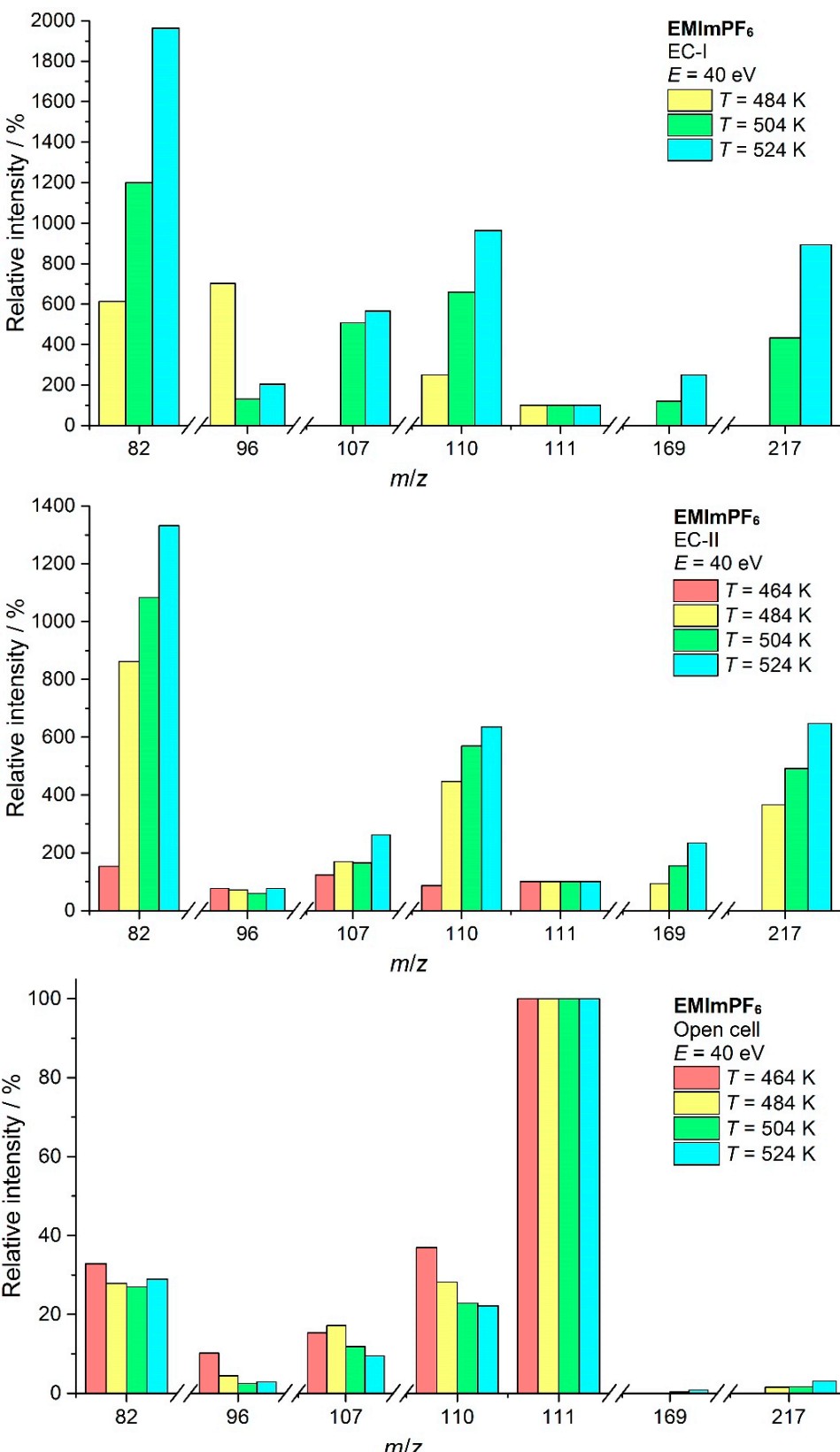

**Figure 6.** *Cont.*

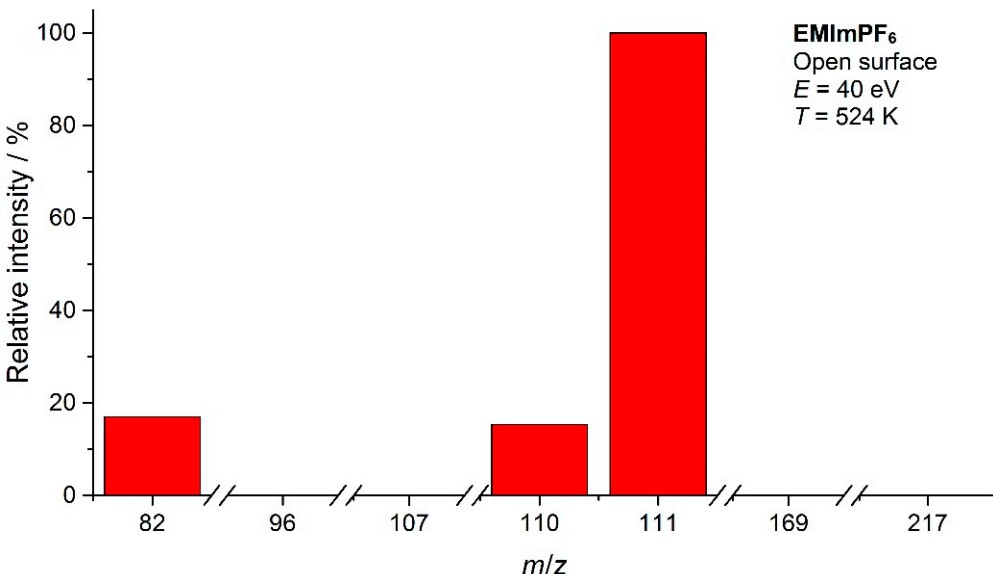

**Figure 6.** Mass spectra of EMImPF$_6$ under different conditions (intensity of the peak with $m/z = 111$ was taken as 100%).

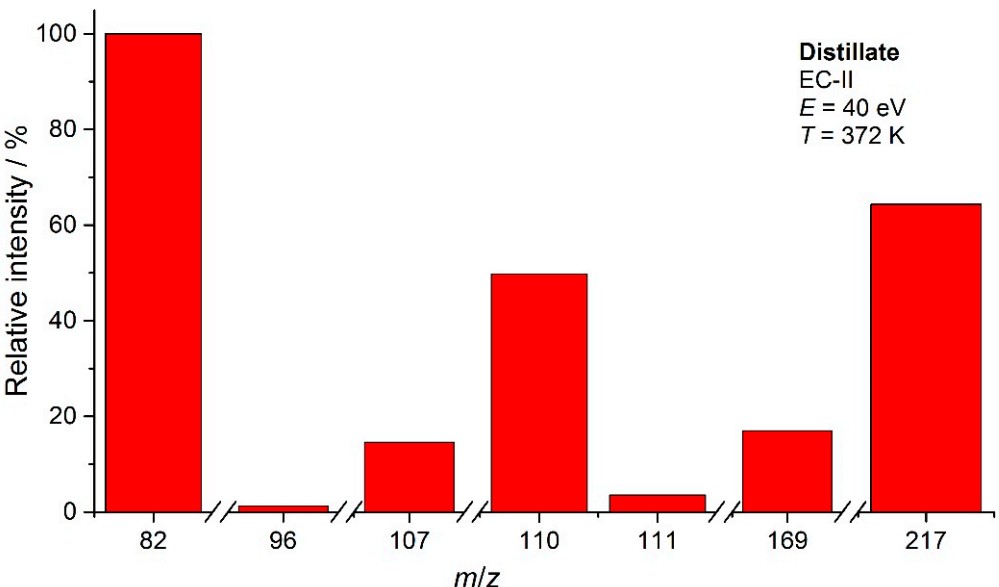

**Figure 7.** Mass spectrum of distillate at $T = 382$ K and $E = 40$ eV (intensity of the peak with $m/z = 82$ was taken as 100%).

The *AE* values of the ions with $m/z = 82$ and 110 under both EC-I and EC-II conditions are temperature-independent. Comparison of their *AEs* between EC and OC reveals a slight decreasing tendency against temperature. Both of these ions are also present in the OS mass spectrum and can originate from NIPs; the low intensity of the ion currents did not allow measuring their IECs under OS conditions. The values $AE(82) = 9.9 \pm 0.3$ eV and $AE(110) = 8.6 \pm 0.1$ eV, being the average among EC-I, EC-II, OC (464 K), and OC (484 K), decrease by 2.0 and 0.6 eV, respectively, under OC (524 K) conditions. This behavior can be explained as follows: Under equilibrium conditions, the major contribution into the current of these ions is due to decomposition products. The transition from the Knudsen conditions to the Langmuir conditions is accompanied by a decrease in the amount of decomposition products; therefore, the contribution from NIPs becomes greater leading to a decreasing of the *AE* values. The *AE* values of the ions with $m/z = 130$ and 367 under OC conditions at 524 K are $10.3 \pm 0.5$ eV and $10.9 \pm 0.5$ eV, respectively.

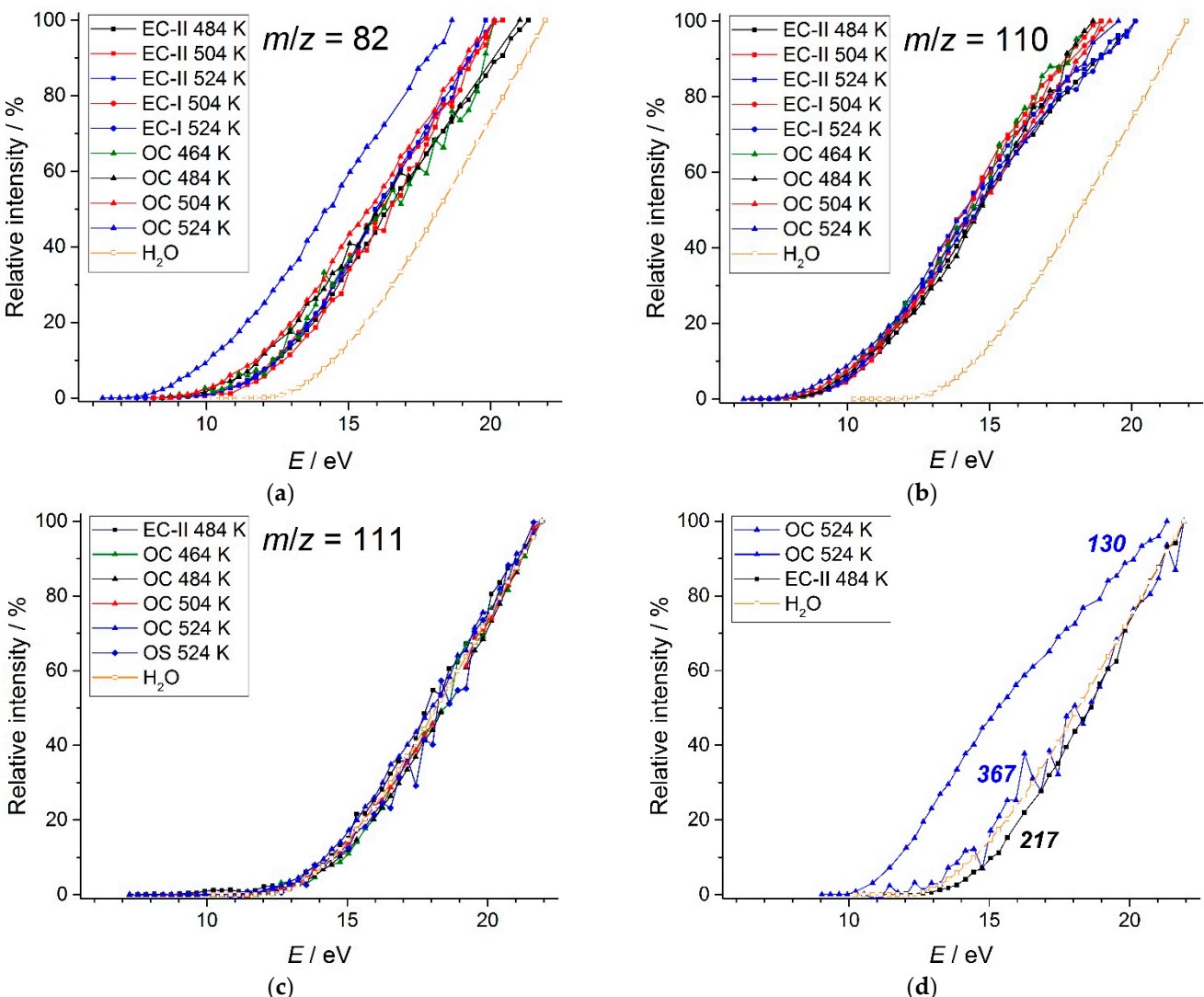

**Figure 8.** Ionization efficiency curves of ions with $m/z$ = 82 (**a**), 110 (**b**), 111 (**c**), and others (**d**) in mass spectra of EMImPF$_6$ under different evaporation conditions.

IECs for the IL distillate measured under EC-II conditions at 374 K are shown in Supplementary Materials (Figure S6). *AEs* of the main ions are collected in Table 1. One can see that they are close to those obtained on equilibrium conditions, confirming our suggestion that the C$_6$N$_2$H$_{10}$PF$_5$ is the main molecular precursor of the ions with $m/z$ = 82 and 110. The $AE$(217) = 13.7 ± 0.5 eV for the distillate agrees with that measured for IL under EC-II conditions at 484 K (13.1 ± 0.5 eV).

The temperature dependencies of the ion currents were measured under different evaporation conditions and their slope tangents are collected in Table 2. One can see that the slope of the temperature dependence of the ion with $m/z$ = 111 is near under all conditions. The slopes for ions with $m/z$ = 110, 111, and 130 under OC and EC-II conditions are very close to each other, indicating NIPs as the common source of their origination. The slope for an ion with $m/z$ = 217 corresponding to the detachment of fluorine from C$_6$N$_2$H$_{10}$PF$_5$ is rather high (−16.4). As expected, the ions with $m/z$ = 82 and 110 have two sources of origination: NIPs and C$_6$N$_2$H$_{10}$PF$_5$. Hence, their slopes are intermediate values as shown in Table 2.

To identify the molecular precursors of the observed ions, the dependencies of the relative ion currents on the evaporation conditions are studied (Figure 9). One can see that the difference between the EC-I and EC-II conditions is negligible. The absence of a trend of the 130/111 ratio confirmed the suggestion that NIP is their common source. The same

situation occurs for the 169/217 ratio, pointing out the formation of both these ions from $C_6N_2H_{10}PF_5$. An ion with $m/z = 110$ under equilibrium conditions is primarily formed from substituted imidazole-2-ylidene, while, under nonequilibrium conditions, NIP is its main source. The main change in the considered ratios is observed at the transition from equilibrium (EC-I and EC-II) to nonequilibrium and intermediate (OC and OS) conditions. One can see that the ratios 82/111 and 110/111 are very close for the OC and OS conditions, indicating that, in OC conditions, the intensities of these ions are primarily determined by the contribution from NIPs.

**Table 1.** Appearance energies (*AE*, ±0.5 eV) of main ions at different evaporation conditions.

| Conditions | *T*/K | *m/z* | | |
|---|---|---|---|---|
| | | **82** | **110** | **111** |
| | | IL | | |
| EC-I | 504 | 10.0 | 8.5 | |
| | 524 | 10.0 | 8.5 | |
| EC-II | 484 | 9.8 | 8.8 | 12.1 |
| | 504 | 10.2 | 8.8 | |
| | 524 | 10.1 | 8.6 | |
| OC | 464 | 9.5 | 8.6 | 12.4 |
| | 484 | 9.5 | 8.6 | 12.4 |
| | 504 | 9.4 | 8.3 | 12.3 |
| | 524 | 7.9 | 8.0 | 12.0 |
| OS | 524 | | | 11.9 |
| | | Distillate | | |
| EC-II | 374 | 11.4 | 9.6 | |

**Table 2.** Slope tangents of temperature dependence of ion currents at different evaporation conditions.

| Conditions | EC-I | EC-II | OC |
|---|---|---|---|
| Δ*T*/K | 474–511 | 463–523 | 454–525 |
| *m/z* | | | |
| 82 | −16.457 ± 0.763 | −17.349 ± 0.498 | −15.240 ± 0.379 |
| 110 | −15.636 ± 0.544 | −14.732 ± 0.375 | −14.702 ± 0.232 |
| 111 | −14.049 ± 0.227 | −14.910 ± 0.209 | −14.634 ± 0.245 |
| 130 | | | −14.242 ± 0.234 |
| 217 | −16.839 ± 1.097 | −16.248 ± 0.943 | |

Based on the above conclusions, the following scheme of fragmentation of the vapor species is suggested (Figure 10). Some ions having only one molecular precursor are fingerprints of NIPs ($m/z = 111$ and 130) and $C_6N_2H_{10}PF_5$ ($m/z = 169$ and 217), while the others have two precursors.

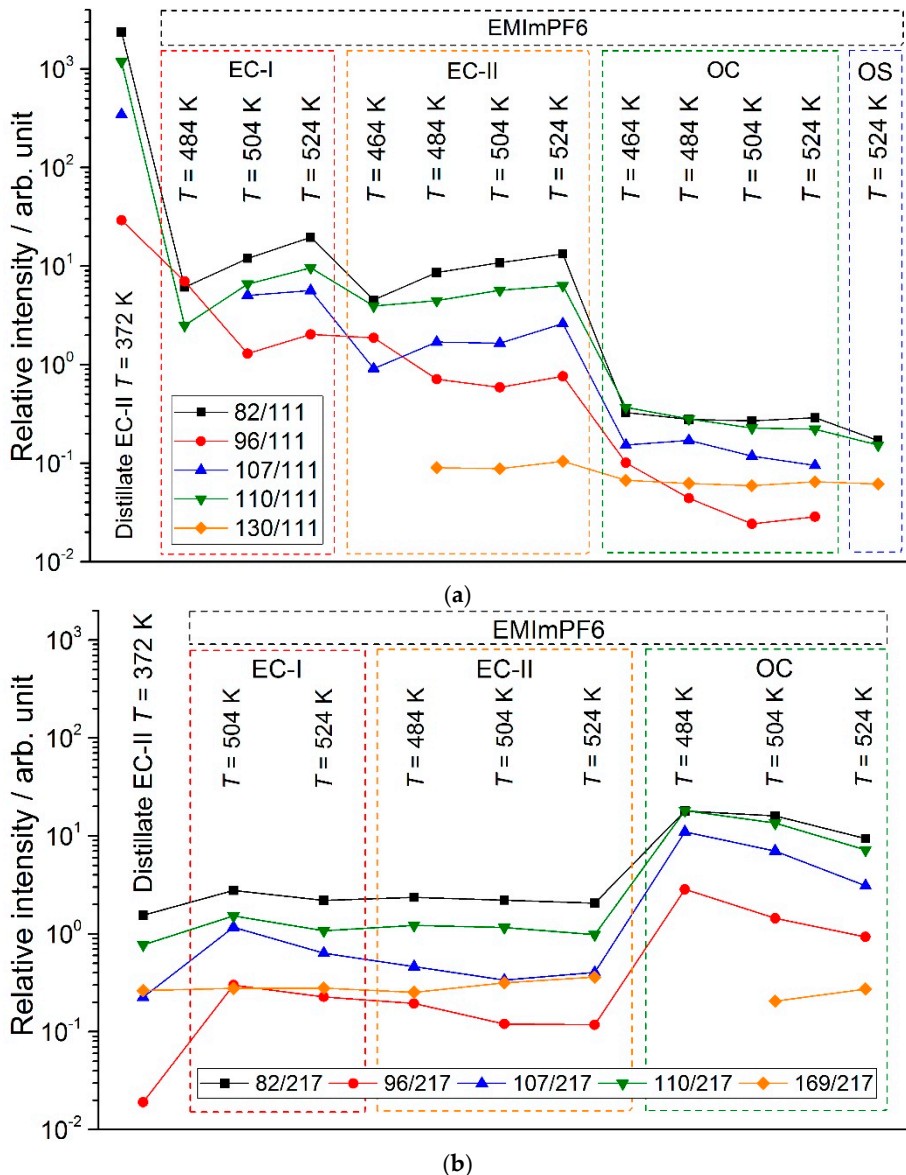

**Figure 9.** Dependence of relative intensity of ion currents on evaporation conditions: (**a**)—relative to $m/z = 111$; and (**b**)—relative to $m/z = 217$.

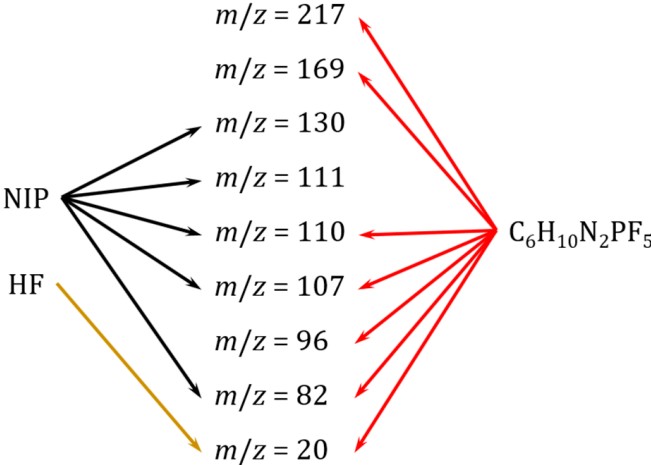

**Figure 10.** Scheme of fragmentation of the vapor species of EMImPF$_6$.

The quantitative interpretation of the mass spectra was performed on a basis of the fragmentation coefficients derived from the OS mass spectrum of IL (NIP is the only molecular precursor) and the spectrum of the distillate ($C_6N_2H_{10}PF_5$ is the only molecular precursor). The contributions into the ion intensities from different sources were found by the following equations:

$$I(82/NIPs) = k(82/NIPs) \cdot I(111), \tag{1}$$

$$I(110/NIPs) = k(110/NIPs) \cdot I(111), \tag{2}$$

$$I(130/NIPs) = k(130/NIPs) \cdot I(111), \tag{3}$$

$$I(82/C_6N_2H_{10}PF_5) = k(82/C_6N_2H_{10}PF_5) \cdot I(217), \tag{4}$$

$$I(96/C_6N_2H_{10}PF_5) = k(96/C_6N_2H_{10}PF_5) \cdot I(217), \tag{5}$$

$$I(107/C_6N_2H_{10}PF_5) = k(107/C_6N_2H_{10}PF_5) \cdot I(217), \tag{6}$$

$$I(110/C_6N_2H_{10}PF_5) = k(110/C_6N_2H_{10}PF_5) \cdot I(217). \tag{7}$$

The fractions of the vapor components (Figure 11) were calculated. One can see from Figure 8 that the vapor composition is almost temperature-independent. Under equilibrium conditions, the substituted imidazole-2-ylidene $C_6N_2H_{10}PF_5$ dominates. The transition to Langmuir conditions leads to a significant increase of the NIP fraction in vapor.

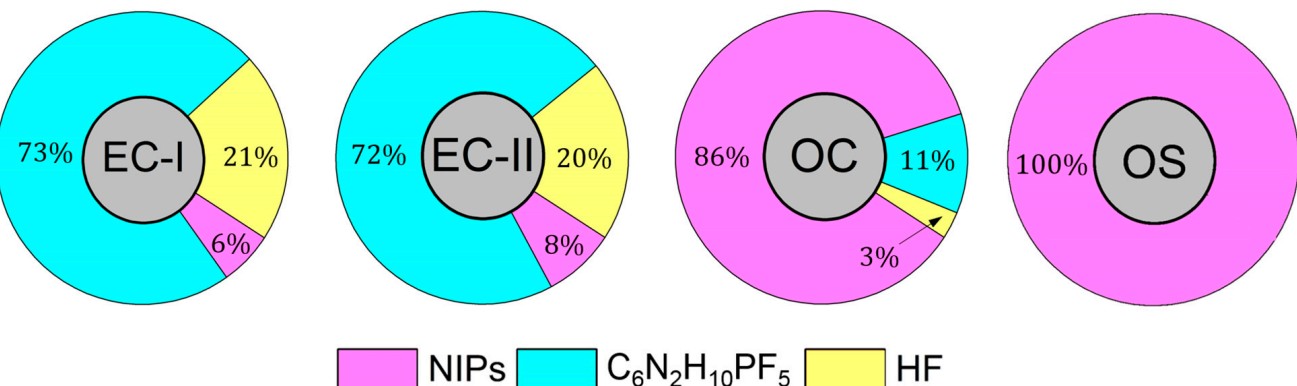

**Figure 11.** Fractions of the vapor components under different evaporation conditions.

The value of the thermodynamic activity of IL is an important question when performing thermodynamic calculations of sublimation enthalpy. Decomposition products are highly volatile and do not accumulate in the condensed phase as shown by the NMR and IR spectroscopy of the IL residue after KEMS (see above). Hence, the IL composition is practically unchanged and its activity is close to unity.

The partial pressures of the vapor species in the Knudsen cell were obtained using their fractions (Figure 11) and the total vapor pressure found by the Knudsen effusion method. The procedure was described earlier [21]. The experiment was carried out at $T$ = 540 K for 19 h using EC-I. The effective area of the effusion orifice, $s$ = (3.40 ± 0.66)·$10^{-8}$ m², was determined by the sublimation of pure zinc; see [21] for details. The average molar mass of the EMImPF₆ IL vapor, $M$ = 191.9 kg·kmol⁻¹, was calculated from the molar masses of the effusion products and their fractions (Figure 11). The found total vapor pressure (0.67 Pa) was used to obtain the sensitivity constant of the mass spectrometer. The experimentally measured ion currents were recalculated into the partial vapor pressures (Table 3) according to a conventional mass spectrometric procedure described elsewhere in details [19]. The temperature dependencies of the vapor pressures (Figure 12) in the form $\ln(p) = f(1000/T)$ were approximated by a linear equation $\ln(p/\text{Pa}) = a \cdot 10^3/T + b$ with the $a$ and $b$ coefficients

given in Table 4. One can see from Figure 11 that the vapor pressure of NIPs obtained in this work is close to that from [7].

**Table 3.** Partial pressures ($\cdot 10^4$ Pa) of vapor species over EMImPF$_6$ under EC conditions.

| $T/K$ | NIPs | C$_6$N$_2$H$_{10}$PF$_5$ | HF |
|---|---|---|---|
| 524 | 16.0 | 157 | 45.6 |
| 514 | 10.3 | 117 | 34.0 |
| 505 | 6.60 | 79.0 | 23.0 |
| 494 | 3.51 | 30.9 | 8.99 |
| 484 | 1.73 | 16.1 | 4.70 |
| 474 | 0.93 | 6.93 | 2.02 |
| 464 | 0.37 | 4.84 | 1.41 |
| 469 | 0.63 | 3.92 | 1.14 |
| 479 | 1.30 | 8.01 | 2.33 |
| 488 | 2.51 | 17.3 | 5.04 |
| 499 | 4.44 | 41.6 | 12.1 |

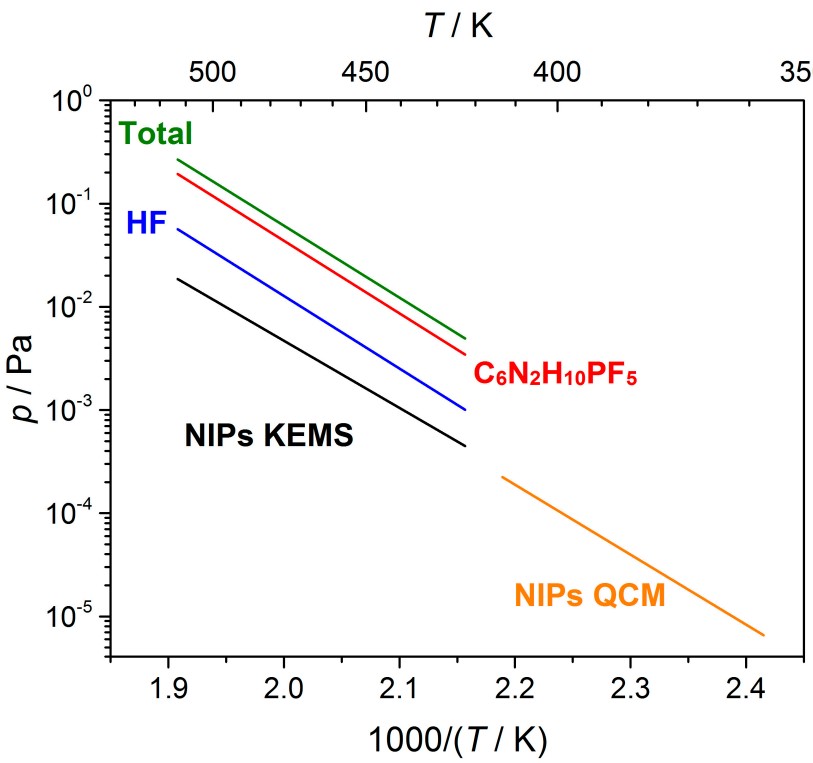

**Figure 12.** Temperature dependencies of pressure of the vapor species over EMImPF$_6$. NIPs QCM data are taken from [7].

The thermodynamic functions of the EMImPF$_6$ IL in the condensed state needed for thermochemical calculations were obtained from the temperature dependencies of the molar heat capacity. The available literature data [22–24] on heat capacity are restricted to $T$ = 355 K and have only four points above melting. Since the heat capacity of liquids usually poorly depends on temperature, the $c_p°$ values above experimental temperatures were taken as a constant equal to $c_p°$(355 K) = 374 J·mol$^{-1}$·K$^{-1}$. $S°$(EMImPF$_{6,cr}$, 298.15 K) = 353 J·mol$^{-1}$·K$^{-1}$ was assessed by the method proposed by Glasser and Jenkins [25]. The density of EMImPF$_6$ in the condensed state needed for the entropy assessment was taken from [26]. The thermodynamic

functions $H°(T)–H°(298.15)$ found in this way are in good agreement with those obtained by an alternative method given in [7] based on the thermal expansion coefficient, isothermal compressibility, and molar volume. The thermodynamic functions used in this work were approximated by polynomial

$$\phi'°(T) = \frac{G°(T) - [H°(T) - H°(298.15\ \text{K})]}{T} = a\ln x + bx^{-2} + cx^{-1} + dx + ex^2 + fx^2, (x = T/1000), \quad (8)$$

the coefficients of which are given in Table 5.

**Table 4.** Coefficients of the equation $\ln(p/\text{Pa}) = a \cdot 10^3/T + b$.

| Vapor Species | *a* | *b* |
|---|---|---|
| *Ionic liquid (464–524 K)* | | |
| NIPs | −14.977 ± 0.380 | 24.592 ± 0.776 |
| $C_6N_2H_{10}PF_5$ | −16.184 ± 0.867 | 29.239 ± 1.772 |
| HF | −16.184 ± 0.867 | 28.005 ± 1.772 |
| *Distillate (332–374 K)* | | |
| $C_6N_2H_{10}PF_5$ | −10.384 ± 0.166 | 25.892 ± 0.475 |

**Table 5.** Coefficients of polynomial (8).

| Compound | $\Delta T/\text{K}$ | *a* | *b*, $\cdot 10^{-3}$ | *c* | *d*, $\cdot 10^3$ | *e*, $\cdot 10^4$ | *f*, $\cdot 10^4$ |
|---|---|---|---|---|---|---|---|
| $EMImPF_{6,\,s}$ | 298.15–334 | 1998 | −1035 | 170.5 | 154.2 | −240.3 | 1340 |
| $EMImPF_{6,\,s}$ | 334–600 | 9.847 | −8.787 | 5.077 | 8.156 | −2.304 | −5.235 |
| $EMImPF_{6,\,g}$ | 298.15–600 | −74.95 | 7.229 | 2.427 | 7.009 | −2.240 | 4.391 |
| $C_6N_2H_{10}PF_{5,\,g}$ | 298.15–600 | −63.76 | 5.581 | 2.336 | 6.232 | −1.911 | 3.562 |

The sublimation enthalpies (Table 6) found in this work by the methods of the second and third law (see Table S2) agree with each other and with the values from [7] as well. Taking into account large uncertainties of the thermodynamic functions, the second-law data are preferable. The value 298.15 K) = 157 ± 6 kJ·mol$^{-1}$, being an average between our data and those from [7], was recommended. Using this value and $\Delta_f H°(EMImPF_{6,\,s}$, 298.15 K) = −2098.9 ± 4.7 kJ·mol$^{-1}$ measured by solution calorimetry [7], the formation enthalpy of $EMImPF_6$ NIPs was found as –1942 ± 8 kJ·mol$^{-1}$.

**Table 6.** Reaction enthalpies $\Delta_r H°$ (kJ·mol$^{-1}$) and entropies $\Delta_r S°$ (J·mol$^{-1}$·K$^{-1}$), temperature range $\Delta T$ (K), mean harmonic temperature $T$ (K), and number of measurements $N$.

| $\Delta T$ | $T$ | $N$ | $\Delta_r H°(T)$ | $\Delta_r H°(298.15\ \text{K})$ | | $\Delta_r S°(T)$ | | Method [3] | Conditions | Ref. |
|---|---|---|---|---|---|---|---|---|---|---|
| | | | | II Law [1] | III Law [2] | II Law [1] | III Law [2] | | | |
| | | | | $EMImPF_{6,\,s,l} = EMImPF_{6,\,g}$ | | | | | | |
| 464–524 | 490 | 11 | 125 ± 3 | 156 ± 4 | 152 ± 14 | 109 ± 4 | 102 ± 18 | KEMS | EC-II | this work |
| 414–457 | 436 | 18 | 130 ± 1 | 158 ± 2 | 153 ± 14 | 118 ± 1 | 108 ± 18 | QCM | OS | [7] |
| | | | | $EMImPF_{6,\,g} = C_6N_2H_{10}PF_{5,\,g} + HF_{,\,g}$ | | | | | | |
| 464–524 | 490 | 11 | 145 ± 8 | 144 ± 9 | 112 ± 14 | 176 ± 24 | 110 ± 18 | KEMS | EC-II | this work |

[1] The standard deviations are given for $\Delta_r H°(T)$. When $\Delta_r H°(T)$ were recalculated to 298.15 K, their uncertainties were increased by those in thermodynamic functions estimated as 2% from $H°(T)–H°(298.15)$ of NIP [16]; [2] Uncertainty is mainly determined by that in thermodynamic functions estimated as 2% from $\phi'(T)$ of NIP [16] and 6% from $S°(T)$ of IL [25]; [3] Methods: Knudsen effusion mass spectrometry (KEMS) and quartz micro balance (QCM).

$\Delta_f H°(EMImPF_{6,\,g}$, 298.15 K) was also calculated by the isodesmic reaction approach with the use of the composite G4 method. The required experimental data on the formation enthalpies

of the reactants are taken from [27,28]. A full list of reactions used as well as the obtained formation enthalpies can be found in Supplementary Materials (Table S1). All values were treated by the Student's method, and the average $\Delta_f H°(EMImPF_{6, g}, 298.15 K) = -1937 \pm 3$ kJ·mol$^{-1}$ was evaluated. This enthalpy is in good agreement with the experimental one.

The enthalpy of the gas-phase reaction (I) was calculated using experimentally determined partial pressures by the method of the second and third law (Table 6 and Figure S7). One can see large discrepancies in the obtained enthalpies and entropies. A theoretical calculation of the reaction enthalpy using the G4 method gives $\Delta_r H°(298.15 K) = 75$ kJ·mol$^{-1}$, which is much lower than both experimental values. It means that the experimental partial pressures of the decomposition products do not reach their equilibrium values. This phenomenon can be explained as the partial equilibrium caused by the kinetically hindered decomposition. Such behavior was previously observed for the similar BMImPF$_6$ IL [3,4].

To test the kinetic factor influence, the time dependencies of the intensity of the ions with $m/z = 111$ (corresponding to NIPs) and $m/z = 110$ (corresponding to $C_6N_2H_{10}PF_5$) were measured at cooling and heating between 484 and 504 K under EC-II conditions with a 20 s step (the heating rate is 5 K/min). The relative gain/loss of intensity were calculated according to the following formula:

$$\Delta I_i = \frac{|I_i - I_0|}{I_{max}} \times 100\%, \tag{9}$$

where $I_0$ is the ion current at the starting temperature, and $I_{max}$ the maximal ion current.

The $\Delta I$ of an ion with $m/z = 111$ rises and falls faster than that with $m/z = 110$ (Figure 13). A rate of the intensity changes was found as a first derivative of the time dependence of the relative ion current. The rate of reaching a steady state for an ion with $m/z = 111$ is higher than that with $m/z = 110$ both at heating and cooling. This fact indicates that the attainment of equilibrium in decomposition reaction (I) is kinetically hindered and any thermodynamic calculations for the decomposition products ($C_6N_2H_{10}PF_{5, g}$ and HF$_{, g}$) are impossible.

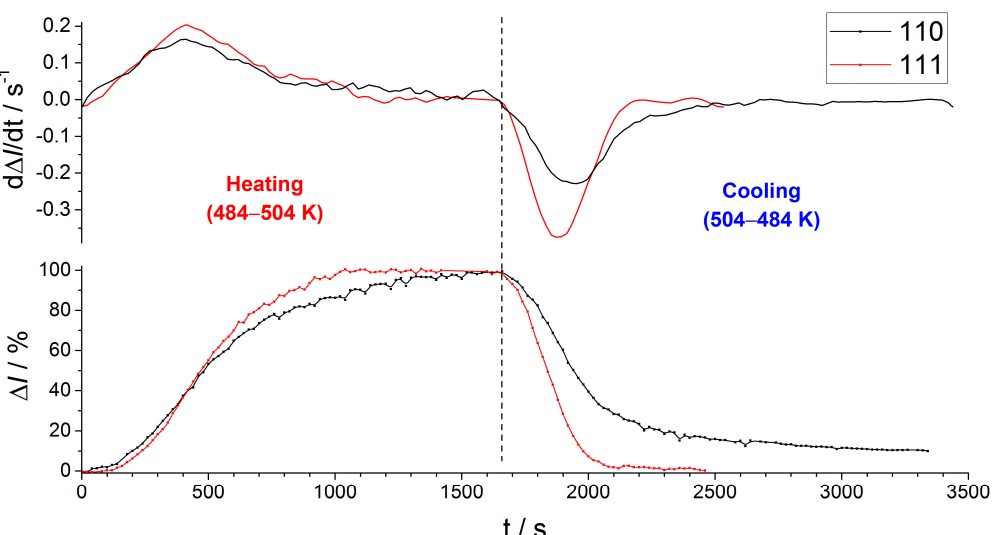

**Figure 13.** Time dependence of the relative gain/loss of intensity of ions with $m/z = 111$ and 110 (down) and its first derivative (up).

The temperature dependencies of the ion currents measured for the IL distillate (90% of $C_6N_2H_{10}PF_5$; Figure S8) show that in the temperature range 332–374 K, all ions are characterized by the close slopes indicating a single source of their origination. The determined vapor pressure of $C_6N_2H_{10}PF_5$ was fitted by a linear equation with coefficients given in Table 4. The plot of the temperature dependence of the vapor pressure is given in Supplementary Materials (Figure S9). The assessed activity $a(C_6N_2H_{10}PF_5) \leq 2 \cdot 10^{-4}$

at an average temperature of the KEMS study of EMImPF$_6$ confirms our assumption that the activity of IL is close to 1. The upper limit of the assessed C$_6$N$_2$H$_{10}$PF$_5$ activity is determined by the fact that the studied distillate is not pure substituted imidazole-2-yidene. The sublimation enthalpy and entropy of C$_6$N$_2$H$_{10}$PF$_5$ at $T$ = 350 K were 86 $\pm$ 1 kJ·mol$^{-1}$ and 120 $\pm$ 4 J·mol$^{-1}$·K$^{-1}$, respectively. The value of sublimation enthalpy is close to those of substituted alkylimidazoles [29]. The enthalpy of the C$_6$N$_2$H$_{10}$PF$_{5,\,g}$ formation at 298.15 K (–1594 $\pm$ 6 kJ·mol$^{-1}$) was calculated from the theoretical enthalpy of the reaction II and the found value $\Delta_f H°$(EMImPF$_{6,\,g}$, 298.15 K).

## 4. Conclusions

The heating of EMImPF$_6$ IL is accompanied by competing vaporization and decomposition processes. The evaporation-to-effusion ratio is a key factor determining the relationship between them. It was established that, under Langmuir conditions, the vapor over IL consists of NIPs only, whereas, under equilibrium conditions, in the temperature range (464–524 K), the decomposition products are dominating. The latter are highly volatile and do not accumulate in a significant amount in the condensed phase. To identify their nature, the IL distillate was collected and analyzed. The analysis provided by the IR and NMR spectroscopy reveals that the IL distillate is 1-ethyl-3-methylimidazolium-2-pentafluorophosphate (C$_6$N$_2$H$_{10}$PF$_5$). No traces of any imidazoles were found. Therefore, there is only one decomposition path—onto C$_6$N$_2$H$_{10}$PF$_5$ and HF. The vapor pressure of C$_6$N$_2$H$_{10}$PF$_5$ obtained by KEMS is used to assess a value of thermodynamic activity (<10$^{-4}$) of the decomposition products in IL. It was assumed that the vapor pressures of the decomposition products do not attain the equilibrium ones and, thus, the decomposition process is kinetically hindered. This assumption was supported by the measurements of the time dependencies of the ion currents corresponding to NIPs and C$_6$N$_2$H$_{10}$PF$_5$ on the one hand, and by the calculation of the enthalpy of the gas-phase decomposition reaction on the other hand.

The NIP vapor pressure and the sublimation enthalpy of EMImPF$_6$ found in this work (156 $\pm$ 4 kJ·mol$^{-1}$ at 298.15 K) are in agreement with the corresponding literature data [7] found in Langmuir conditions. This fact indicates that the vaporization coefficient of EMImPF$_6$ in the form of NIPs is close to unity; the same conclusion was reached earlier for the similar BMImBF$_4$ IL [3]. Taking into account the melting enthalpy 17.7 kJ·mol$^{-1}$ [23] for the EMImPF$_6$ IL, its sublimation enthalpy is reasonably consistent with the vaporization enthalpy of the thermally stable EMImNTf$_2$ IL ($\Delta_v H°$(298.15 K) = 135.3 $\pm$ 1.3 kJ·mol$^{-1}$ [30]). The combination of the experimental formation enthalpy of the solid EMImPF$_6$ and the obtained sublimation enthalpy gives $\Delta_f H°$(EMImPF$_{6,\,g}$, 298.15 K) = –1942 $\pm$ 8 kJ·mol$^{-1}$, being in good agreement with that calculated using the composite G4 method.

From the results obtained, it can be practically concluded that, in physical vapor deposition technology, where the substance is evaporated from the open crucible, decomposition processes will be negligible over the entire temperature range studied.

**Supplementary Materials:** The supporting information can be downloaded at: https://www.mdpi.com/article/10.3390/appliedchem3020019/s1.

**Author Contributions:** Conceptualization, A.M.D.; methodology, V.B.M.; software, D.G. and V.V.A.; validation, L.S.K.; formal analysis, A.M.D., V.B.M. and M.A.K.; investigation, V.B.M., V.V.A. and M.A.K.; resources, D.G.; writing—original draft preparation, A.M.D.; writing—review and editing, V.B.M.; visualization, A.M.D.; supervision, L.S.K.; project administration, A.M.D.; funding acquisition, A.M.D. All authors have read and agreed to the published version of the manuscript.

**Funding:** This research was funded by Russian Science Foundation under grant No. 21-73-00041. The study was carried out using the resources of the Center for Shared Use of Scientific Equipment of the ISUCT (with the support of the Ministry of Science and Higher Education of Russia, grant No. 075-15-2021-671).

**Data Availability Statement:** Data supporting this study are openly available from the Manuscript and Supplementary Materials.

**Conflicts of Interest:** The authors declare no conflict of interest.

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
