# Peer review of "Vapor Composition and Vaporization Thermodynamics of 1-Ethyl-3-methylimidazolium Hexafluorophosphate Ionic Liquid"

_appliedchem, doi:10.3390/appliedchem3020019_

Round 1

Author Response

Reviewer 1

I am very grateful you for the invitation to review the manuscript appliedchem by Dunaev and coauthors “Vapor composition and vaporization thermodynamics of 1‐ethyl‐3‐methylimidazolium hexafluorophosphate ionic liquid”. The work is interesting but needs adjustments to increase the quality of the material.

Comments:

  • The authors should report the water content in the IL.

We assume that the water content is negligible. The new commercial samples of IL under study were kept in the desiccator to avoid any influence of moisture. Sampling procedure was provided in a relatively short time. The sample pretreatment was performed under forevacuum in accordance to the standard procedure used in our experimental setup to remove absorbed moisture from the sample before transition to high vacuum and heating the sample to the evaporation temperatures. So the temperature is always selected equal to or slightly over 373 K (water boiling point) to provide effective desorption on the one hand and to avoid interaction with the sample on the other hand. Usually 1.5-2 hours is enough to achieve a stable residual pressure inside the unit, which indicates the end of water desorption. It should be also noted that the water signal in NMR spectra can appear from water absorbed by DMSO-d6.

  • It would be of interest to calculate the impurity found in the IL using the NMR spectra.

The NMR analysis is an inappropriate method for quantitative assessment of the impurities. This is due to the fact that if the structure of impurities is not known in advance, it is impossible to determine the number of protons in the desired signal and, therefore, to carry out the integration. In addition, the uncertainty of the integration (about 5%) cannot allow to perform reliable calculation of signals comparable to noise level.

  • More than 50% of the references are outdated. Please add recent references and discuss them.

We fully disagree with this non-specific comment. Among references there are only 3 older than 2000 and 4 older than 2010. It is obviously less than 50% of the reference list. 7 references are from the last five years (2018-2023).

  • Do the studied IL ions remain bound or are they dissociated at high temperatures?

The studied IL in the liquid phase is dissociated, but in the gas phase it presents in the form of NIP in a whole temperature range, as was established earlier [Armstrong, 2007]. The amount of ions in the vapor may be considered negligible [Dunaev, 2016]

Reviewer 2 Report

The authors studied the vaporization and decomposition of EMImPF6 with analysis including IR, NMR and mass spectrometry. In general, this is an interesting study. However, there are still some questions needed to be answered and/or issues needed to be addressed before being published:

1.    For Figure 5, EMImPF6 under EC conditions, why there are 4 temperatures for EC-II while only 3 temperatures for EC-I? Meanwhile, would the authors help me to better understand the reasons for using different ions (m/z=111 for Figure 5 and m/z=82 for Figure 6) to calculate the relative intensities?

2.      As the authors mentioned, their previous study involving another ionic liquid BMImBF4. It seems these two studies are similar to each other. I wonder what the novelty points are compared with previous study? I would recommend the authors discuss some more about the novelties of this study.

Author Response

Reviewer 2

The authors studied the vaporization and decomposition of EMImPF6 with analysis including IR, NMR and mass spectrometry. In general, this is an interesting study. However, there are still some questions needed to be answered and/or issues needed to be addressed before being published:

  1.   For Figure 5, EMImPF6 under EC conditions, why there are 4 temperatures for EC-II while only 3 temperatures for EC-I? Meanwhile, would the authors help me to better understand the reasons for using different ions (m/z=111 for Figure 5 and m/z=82 for Figure 6) to calculate the relative intensities?

The intensities of the ion currents under EC-I conditions are too low to be measured at T = 464 K. This is due to the smaller size of the effusion orifice in the EC-I in comparison to EC-II.

The mass spectra of the IL are shown in Fig. 5, while in Fig. 6 the spectrum of decomposition products. In the first case the parent cation m/z = 111 is a “fingerprint” of NIP and all ions were normalized by it. In the case of distillate the amount of IL is negligible, the mass spectrum corresponds to C6N2H10PF5 and all intensities were recalculated relative to the intensity of the main ion with m/z = 82.

  1. As the authors mentioned, their previous study involving another ionic liquid BMImBF4. It seems these two studies are similar to each other. I wonder what the novelty points are compared with previous study? I would recommend the authors discuss some more about the novelties of this study.

As was mentioned in the manuscript we apply the same approach discussed in our previous study for 1‐ethyl‐3‐methylimidazolium hexafluorophosphate (EMImPF6) being a representative of alkyl-imidazolium ILs with the PF6 anion. This class of IL was not investigated before by this approach. In addition, in this research the temporal dependencies of ion currents were studied, which confirmed the kinetically hindered decomposition.

Reviewer 3 Report

The Authors submitted an interesting study on the competition between vaporization and decomposition of a well-know Ionic Liquid (IL). It would be a welcome addition to the existing studies on the subject. In my opinion, the manuscript would benefit greatly from details on the mechanism of the decomposition reaction. Furthermore, a clear statement is missing where decomposition would occur, in the liquid, the gas phase or at the interface.

As a practical consequence of the study for the (technical) application of the results, I suggest the Authors provide a summary in which range this IL can be deposited on a substrate via physical vapor deposition (PVD) without decomposition, e.g. for the preparation of thin films, and under which conditions decomposition would accompany PVD.
Earle et al. [Nature 2006, 439 (7078), 831] already noted that decomposition in alkylimidazolium PF6 ILs can occur upon destillation.
"(EMImPF6) being a representative of alkyl-imidazolium ILs with the PF6– anion. This IL is expected to decompose according to reaction..."
How do the authors come to this expectation? Reference to literature needed?
"1-ethyl-3-methylimidazolium-2-pentafluorophosphate" Is this unambiguously supported by the data? Is this a stable compund? Intuitively, I find it hard to believe, the bulky PF5 could bind directly to the C2. The authors should provide more background. Could we expect the formation of C6N2H10 + HPF6 (--> deprotonation of the cation) or C6N2H10-F + PF5?
What could other decomposition products be, that the Authors were unable to distill and analyze? Could these be relevant for applications of this IL at elevated temperatures, e.g. in catalysis?

If there is no doubt, the main decomposition products are C6N2H10PF5 and HF, it would benefit the readers greatly, if the authors would suggest a reaction mechanism and provide the structure formula of the resulting C6N2H10PF5 molecule.
Is there any way to tell, if the decomposition occurs in the liquid phase or at the surface? Or in the gas phase for isolated ion pairs or for small clusters of two, several or many ion pairs?
Please check that all abbreviations/acronyms are introduced at first mention, for example: KEMS, ...
Figure 10: The values in the pie charts are for one specific temperature? Or all temperatures?
Figure 12: What is the heating and cooling rate [K/s]?

Generally ok. I noticed a few not so common phrases and mistakes.

Author Response

Reviewer 3

The Authors submitted an interesting study on the competition between vaporization and decomposition of a well-know Ionic Liquid (IL). It would be a welcome addition to the existing studies on the subject. In my opinion, the manuscript would benefit greatly from details on the mechanism of the decomposition reaction. Furthermore, a clear statement is missing where decomposition would occur, in the liquid, the gas phase or at the interface.

  1. As a practical consequence of the study for the (technical) application of the results, I suggest the Authors provide a summary in which range this IL can be deposited on a substrate via physical vapor deposition (PVD) without decomposition, e.g. for the preparation of thin films, and under which conditions decomposition would accompany PVD.

From the results obtained it can be practically concluded that in physical vapor deposition technology, where the substance is evaporated from the open crucible, decomposition processes will be negligible over the entire temperature range studied. This information has been added in the manuscript.

  1. Earle et al. [Nature 2006, 439 (7078), 831] already noted that decomposition in alkylimidazolium PF6 ILs can occur upon destillation.

In the mentioned work of Earle author said “Examples of tosylate ([OTs], [Me-4-C6H4SO3]) and hexafluorophosphate ([PF6]) ionic liquids distilled very slowly, with little decomposition (the [PF6] salt must be free of acidic or basic impurities for a clean distillation)”. There are no specifications of the decomposition products, as well as thermodynamic calculations in this paper. Therefore we excluded it from consideration.

  1. "(EMImPF6) being a representative of alkyl-imidazolium ILs with the PF6– anion. This IL is expected to decompose according to reaction..." How do the authors come to this expectation? Reference to literature needed?

As was written in beginning of the manuscript “Vaporization of ionic liquids (ILs) with highly electronegative anions under high vacuum conditions is accompanied by partial decomposition of ILs with the formation of substituted imidazole‐2‐ylidenes. In particular, such decomposition products were clearly identified for ILs with the BF4 [1-3], PF6 [4], dicyanamide, thiocyanate, tricyanomethanide, and vinylogous dicyanamide anions [5].”. The given reaction is discussed in [4], we add the required citation.

  1. "1-ethyl-3-methylimidazolium-2-pentafluorophosphate" Is this unambiguously supported by the data? Is this a stable compund? Intuitively, I find it hard to believe, the bulky PF5 could bind directly to the C2. The authors should provide more background. Could we expect the formation of C6N2H10 + HPF6 (--> deprotonation of the cation) or C6N2H10-F + PF5?

The formation of 1-ethyl-3-methylimidazolium-2-pentafluorophosphate is clearly confirmed by NMR, IR and KEMS analysis (see manuscript). Also the similar compounds were separated by [Taylor, 2011]. We do not provide the investigation of the thermal stability of the C6N2H10PF5, but it is stable on air and under visible light. The formation of imidazoles as well as C6N2H10F is highly unlikely event, because in the NMR spectra there are no signals attributed to them.

  1. What could other decomposition products be, that the Authors were unable to distill and analyze? Could these be relevant for applications of this IL at elevated temperatures, e.g. in catalysis?

According to the NIST database among decomposition products (including imidazoles) the only HF unable to distill at room temperature (HF boiling point is 19.5 ºC). Therefore, all decomposition products were collected and analyzed.

  1. If there is no doubt, the main decomposition products are C6N2H10PF5 and HF, it would benefit the readers greatly, if the authors would suggest a reaction mechanism and provide the structure formula of the resulting C6N2H10PF5 molecule.

The structural formula of C6N2H10PF5 has been added in the text. Our investigation is devoted to the interpretation of the vapor composition and thermodynamic calculations. The study of reaction mechanism is a special task. However, this mechanism for similar reaction between alkylimidazolium cation and BF4 anion can be found in [Taylor, 2011]: borane-substituted imidazol-2-ylidenes have been prepared by the direct nucleophilic reaction of the preformed Arduengo type N-heterocyclic carbene (NHC), with a strongly electrophilic trap, in this case BF3.

  1. Is there any way to tell, if the decomposition occurs in the liquid phase or at the surface? Or in the gas phase for isolated ion pairs or for small clusters of two, several or many ion pairs?

We consider that there are no collisions and reactions among molecules in the gas phase, because of volume concentrations of gaseous species are about 1020 m-3 in our conditions. The question: where the decomposition occurs in the liquid phase or at the surface is still open. However, we assume that it is a process occurring on the surface due to its excess energy.

  1. Please check that all abbreviations/acronyms are introduced at first mention, for example: KEMS, ...

It has been corrected.

  1. Figure 10: The values in the pie charts are for one specific temperature? Or all temperatures?

Taking into account weak temperature dependence of the vapor composition the fractions in Fig. 10 can correspond to the whole studied temperature range.

  1. Figure 12: What is the heating and cooling rate [K/s]?

As written in the manuscript, the heating/cooling rate is 5 K/min.

Reviewer 4 Report

The Authors describe the evaporation to effusion ratio is a key factor determining the relationship between competing vaporization and decomposition processes by the IR and NMR spectroscopy. The NIP vapor pressure and the sublimation enthalpy of EMImPF6 are in agreement. I think the paper will be of interest to the readership of Applied Sciences and I recommend the paper is accepted with some minor changes. Some specific comments are shown below:

 1.          What is the effect of different temperatures on the structure of 1‐ethyl‐3‐methylimidazolium hexafluorophosphate ionic liquid?

2.          Is the distillate stable? What is the procedure for conducting the experiment?

3.          Can the authors show this structure for example m/z=82 in the Fig6.?

Author Response

Reviewer 4

The Authors describe the evaporation to effusion ratio is a key factor determining the relationship between competing vaporization and decomposition processes by the IR and NMR spectroscopy. The NIP vapor pressure and the sublimation enthalpy of EMImPF6 are in agreement. I think the paper will be of interest to the readership of Applied Sciences and I recommend the paper is accepted with some minor changes. Some specific comments are shown below:

  1. What is the effect of different temperatures on the structure of 1‐ethyl‐3‐methylimidazolium hexafluorophosphate ionic liquid?

The structure of the IL does not depend on temperature.

  1. Is the distillate stable? What is the procedure for conducting the experiment?

We did not investigate the thermal stability of the distillate, but it is stable on air and under visible light. In the mass-spectrometric experiment with distillate, ion currents were reproducible in heating and cooling cycles, which can be seen as a sign of stability. The description of the distillation apparatus and the procedure of the distillate collection is given in the manuscript in Section 2.

  1. Can the authors show this structure for example m/z=82 in the Fig6.?

The determination of the ion structure is out of our research. Its structure is not necessary for the mass spectrum interpretation and further thermodynamic calculations. The odd electron ion with m/z=82 corresponds to C4N2H6+ (similar to methylimidazole). The detailed description of the ions in the mass spectra of such ionic liquids can be found elsewhere [Deyko, 2011]
